# (Un)Heard Voices of Ecosystem Degradation: Stories from the Nexus of Settler-Colonialism and Slow Violence

**Leane Makey** [1,*], **Meg Parsons** [1], **Karen Fisher** [1], **Alyssce Te Huna** [2], **Mina Henare** [2], **Vicky Miru** [2], **Millan Ruka** [3] and **Mikaera Miru** [2]

1   School of Environment, University of Auckland, Auckland 1142, New Zealand
2   Te Uri o Hau, Ngāti Whātua, Matakohe 0953, New Zealand
3   Te Uriroroi, Te Parawhau, Te Māhurehure ki Whatitiri, Ngāpuhi-Nui-Tonu, Porotī 0179, New Zealand
*   Correspondence: leane.makey@auckland.ac.nz

**Abstract:** We examine the ecosystem degradation of the Kaipara moana as an example of the nexus of settler colonialism and slow violence. Settler colonialism is a type of domination that violently interrupts Indigenous people's interactions and relationships with their land-, sea-, and water-scapes. Slow violence provides a conceptual framework to explore the slow and invisible erosion of ecosystems and to make visible how unseen violence inflicted upon nature (such as deforestation and sedimentation pollution) also unfolds at the intimate scale of the Indigenous body and household. Here, we present how the structural violence of settler colonialism and ecological transformations created a form of settler colonial slow violence for humans and more-than-humans which highlights the ethical and justice features of sustainability because of the link with settler-colonialism. We argue for the need to include local knowledge and lived experiences of slow violence to ensure ethical and just ensuring practices that better attend to the relationships between Indigenous peoples and their more-than-human kin (including plants, animals, rivers, mountains, and seas). We build on this argument using auto- and duo-ethnographic research to identify possibilities for making sense of and making visible those forms of harm, loss and dispossession that frequently remain intangible in public, political and academic representations of land-, sea-, and water-scapes. Situated in the Kaipara moana, Aotearoa New Zealand, narratives are rescued from invisibility and representational bias and stories of water pollution, deforestation, institutional racism, species and habitat loss form the narratives of slow violence. (Please see Glossary for translation of Māori language, terms and names.)

**Keywords:** Indigenous Māori; slow violence; eco-social violence; ecosystem restoration; geo-creative practices; knowing-doing; settler-colonialism

## 1. Introduction

Most of us were all there at the meeting being held at the tribal marae when we heard the announcement. The silence echoed around the room. Our tribal elders listened intently; heads bowed. Resource consent approval for 35 years had been granted (by the Northland Regional Council, March 2011) to deploy 200 underwater marine turbines in the Kaipara moana entrance, a place where deep, significant tidal flow occurs. Each turbine would be the size of a four-bedroom house. The Company was developing infrastructure in a significant (ecologically and spiritually) sensitive area and did not have final development plans. One of the tribal elders mumbled "this is bull shit" and words like sacred land and sea. Te Taou tribal representatives were silent. The news was hurting them, knowing the Kaipara, their family member, was suffering intergenerational trauma and crisis. For Te Taou the signs were there. Animals they had not seen for a very long time had appeared in the Kaipara. Other signs appeared in the sky and the soil, and the ground.

The large-scale developments proposed for the Kaipara moana, in most cases, do not align with te ao Māori worldview because they fail to place the Kaipara moana at the

centre. Economic drivers dominate the purpose and vision of development in the Kaipara moana while the agency of the Kaipara itself is often neglected. Notably, the Kaipara moana entrance, which locals call the graveyard, is a location of upheaval, violence and death. The place is wāhi tapu for the Kaipara tribes and is described by the proverb "*Taporapora whakatahui waka, whakarere wahine Taporapora*" (Taporapora that capsizes canoes and bereaves women), a reminder of the many lives lost. It is estimated that over 130 shipwrecks have occurred in this location since European settlement started in the 1840s. Recently, in 2016, the charter fishing boat, Francie, went down when it attempted to cross the Kaipara sand bar, with the loss of 9 of 11 souls. The entrance is dynamic and characterised by the deposition of sand-scapes, and ebbing and flowing movements, which creates an intricate and deceptive array of sandbars. These sandbars shift and move, appear and disappear over time and can create safe navigable channels into the Kaipara. Locals talk about how up to 15 channels can appear, and some are safe, and some deadly. Such accounts of the Kaipara's agency are absent in the meeting, which eventually comes to a close with the recitation of a karakia by tribal elders.

In this paper, we examine the environmental degradation of the Kaipara moana as an example of the nexus of settler colonialism and slow violence. Slow violence provides a conceptual framework to explore the slow and invisible erosion of nature justice [1] that exists under capitalism and neoliberalism (both forms of colonisation) and to make visible how unseen violence inflicted upon nature (such as deforestation and sedimentation pollution) also unfolds at the intimate scale of the Indigenous body and household. In emphasising the environmental injustices experienced by nature and Indigenous peoples because of settler colonialism, we argue the need to include local knowledge and lived experiences of slow (environmental and ecosystem) violence to ensure ethical and just ensuring practices that better attend to the relationships between Indigenous peoples and their more-than-human kin (including plants, animals, rivers, mountains, harbours, seas). Partly informed by Spivak's notion of epistemic violence [2], which draws attention to power asymmetries and politics in knowledge production, we show how Indigenous stories have been marginalised and excluded from conventional/dominate, and usually scientific, accounts of environmental management and degradation. We are particularly interested in exploring how the structural violence of settler colonialism and ecological transformations created a form of settler colonial slow violence for humans and more-than-humans. We build on this argument using auto- and duo-ethnographic research to identify possibilities for making sense of and making visible those forms of harm, loss and dispossession that frequently remain intangible in public, political and academic representations of land-, sea-, and water-scapes.

The structure of this article is as follows. Part 2 reviews the concept of slow violence and its use concerning toxic pollution and degraded environments. Second, we highlight the intersections between slow violence and settler colonialism to demonstrate how the structural violence against Indigenous peoples is irrevocably linked with violence against more-than-humans (including plants, animals, rivers, mountains, harbours, seas) that Indigenous peoples consider to be their kin. Part 3 outlines the empirical case study of the degraded coastal-estuarine ecosystem (Kaipara moana) in northern Aotearoa New Zealand. Part 4 explains the methodological approach taken for this research. Part 5 presents a collection of stories, the result of an ethnographic study with Indigenous Māori women (wahine) and men (tane) living in their ancestral lands and waters of Kaipara moana, which make seen (and heard) the impacts of settler colonisation on human and more-than-human actors. Finally, Part 6 concludes by drawing attention to the different ways in which the stories make seen (and heard) a form of settler colonial slow violence that has and continues to affect humans and more-than-humans.

## 2. Slow Violence and the Eco-Social Violence of Settler Colonialism

Settler colonialism is a type of domination that violently interrupts Indigenous people's interactions with and relationships to their environments [3,4]. It is widely argued that

settler colonialism is a form of ecological domination that results in environmental injustices against Indigenous peoples and other marginalised peoples [5–9]. Eve Tuck and K. Wayne Yang [10] write how "the disruption of Indigenous relationships to land represents a profound epistemic, ontological, cosmological violence". Similarly, Vanessa Watts [11] declares that "the measure of colonial interaction with land has historically been one of violence where land is to be accessed, not learned from or a part of ". J.M. Bacon argues that settler colonial states have deployed numerous "mechanisms of eco-social disruption", which has caused land and water to be "redistributed, privatized, polluted, and renamed with generally no input or consent on the part of the original [Indigenous] inhabitants; the value of places and beings are redefined by the culture of the colonizers". These actions contribute to an array of harms to the physical and emotional health and wellbeing of Indigenous peoples and their more-than-human kin [12].

Settler colonial ways of governing and managing environments results in multiple harms (physical, emotional, economic, spiritual, and cultural) for Indigenous peoples; their capacities to interact with and care for (encompassing their responsibilities to be environmental guardians) their environments being severely restricted by settler colonial laws, land loss, and institutional arrangements that discriminate against Indigenous peoples, knowledges, and cultural practices. Such eco-social disruptions, Bacon suggests, are a form of settler colonial ecological violence, a distinct type of violence perpetrated by the settler colonial state, private businesses, and the broader settler colonial culture [12]. Although some scholars concentrate on genocide and ecocide in their analyses of Indigenous peoples and environmental practices [13–15], we instead focus on settler colonial ecological violence as a term that provides a wider analysis of the plethora of avenues by which settler colonialism disrupts Indigenous eco-social relationships and generates specific harms for Indigenous peoples and their more-than-human kin.

For generations, Indigenous scholars, activists, and leaders have spoken of the primary importance of environments (land, water, biota, and sea) to their identities, livelihoods, and ways of life, and clearly articulated how settler-driven ecological changes resulted in multiple forms of loss and hurt for Indigenous peoples. The eminent Māori academic Linda Tuhiwai Smith [16], for example, describes how settler colonial violence not only involved the use of colonial military forces against Indigenous people, land loss, and the wholesale removal of ecosystems but also occurred through the act of renaming places. Scholars in Australia, Canada and the United States similarly draw attention to how the act of renaming is a form of ecological violence and environmental injustice [17–20]. Accordingly, we deliberately use Māori terms and names throughout this paper to resist the violence and re-assert te reo (Māori language), mātauranga (knowledge), and tikanga (Māori laws) as the first language, knowledge, and laws of Aotearoa (New Zealand), which continue despite settler colonialism.

Environmental degradation is a form of slow violence. As Nixon [21] writes: "By slow violence I mean a violence that occurs gradually and out of sight, a violence of delayed destruction that is dispersed across time and space, an attritional violence that is typically not viewed as violence at all". Slow violence necessitates thinking about what constitutes harm and violence beyond short-term time scales and singular causes to consider "slowly unfolding environmental catastrophes" [21]. Slow violence provides an avenue by which to look beyond the obvious and the immediate in examining environmental changes and social injustices. The spatial dimensions of slow violence allow scholars to contemplate the gradual destruction of ecosystems, damages and losses, and the layered deposition of inequitable social and ecological brutalities within contemporary geographies (the here-and-now of specific places and communities). The temporal dimensions of slow violence allow scholars to delve into the past to understand the drivers of violence including the historic drivers (policies, knowledge, laws, and practices) that created inequitable power structures and mechanisms in the present and may contribute (unless actively challenged) to future injustices and violence in the future.

The impacts of slow violence are often temporally latent (gradually cumulative), camouflaged, and interwoven, which makes identification of slow violence a challenge as well as actions to address it. The delayed violence of water and land pollution, climate change, sedimentation, and toxic residues do not necessarily fit popular portrayals of global environmental change and disasters. Yet, slow violence demonstrates how geographies of violence come into being and continue to operate (past, present, future) because of human activities, and that certain populations are more likely to be negatively impacted than others. Confronting slow violence requires providing narratives and giving representative form to shapeless risks whose dangerous outcomes are spread across time and space [19,20,22]. Those communities who live with the daily realities of slow violence (stretched across generations) are best situated to bear witness to its gradual harms.

In Aotearoa, like in other settler colonial states, settler colonialism was (and still is) distinguished by processes and practices of terraforming (transforming Indigenous landscapes, waterscapes, and seascapes to settlerscapes) [12,23,24]. As Anishinaabe First Nation scholar Kyle Whyte writes, "industrial settler campaigns erase what makes a place ecologically unique in terms of human and nonhuman relations, the ecological history of a place, and the sharing of the environment by different human societies" [25]. A constitutive part of settler colonialism involved Indigenous peoples being dispossessed and displaced, and Indigenous communities being forced to adjust to drastically changed environments, biota, ecosystems, and socio-economic conditions.

Indigenous peoples (to varying degrees depending on their colonial situations) are living in geographies of slow violence [7,26]. As a concept, therefore, slow violence unshackles violence from a specific event or time, and instead allows scholars to examine acts of violence (physical, slow, structural, ontological, and epistemological) as layered and built up onto one another over time. Slow violence captures the diversity of social-ecological violent acts such as how settler colonial legal systems, government policies, and institutions promote land-use practices that pollute and harm environments, and which exclude more-than-human others (who are viewed as members of Indigenous people's extended families or their more-than-human kin) [20,27,28]. In turn, as O'Lear observes [29], slow violence can result from political and epistemic dominance of specific understandings, forms of knowledge, and narratives. Critically, the politics of official or expert indifference about the plight of marginalised peoples help to maintain environmental injustices and allow communities' claims about harms to be silenced by those in positions of authority. Disregarding Indigenous and local communities' claims of environmental injustice can serve to establish and maintain a self-reinforcing cycle of violent brutality that is concurrently epistemic, structural, and slow. Non-scientific knowledge, including Indigenous knowledge and local knowledge, is frequently overlooked in accounts of environmental degradation and risk. For instance, in Aotearoa, Parsons et al. [4] observe that settler colonial governments and courts continue to privilege scientific knowledge over local Māori knowledge and consider scientific expertise the legible or legal form for decision-making as opposed to Māori knowledge and laws. This can create "narrative mismatches" between official (settler colonial government and/or scientific expert) and unofficial (Indigenous and/or local) stories of polluted places [30]. Indeed, toxic geographies typically remain ambiguous, disputed, and persistent due to inequitable power structures and mechanisms, lack of recognition of different knowledges, cultures, beings (humans and more-than-humans) and ways of being [22,31,32].

## 3. Kaipara Moana—A Seascape Enduring Slow Violence

Ngāti Whātua, the Indigenous Māori tribal group (iwi) whose ancestral lands and waters include the Kaipara (Figure 1), hold mana whenua status; this means they are the iwi who possess spiritual and political decision-making authority over the Kaipara moana. Their history is briefly introduced in Figure 2 (Timeline), which highlights their centuries of occupation and ongoing reciprocal relationships with Kaipara nature, including the whenua, awa, and moana. The ontology and epistemologies of Ngāti Whātua, like



other iwi/hapū, are intimately intertwined with te taiao, which rests at the heart of Ngāti Whātua experiences of environmental degradation. Accordingly, it is necessary to provide some background to understand Ngāti Whātua worldviews, which are founded on their whakapapa which directly links them to their taiao.

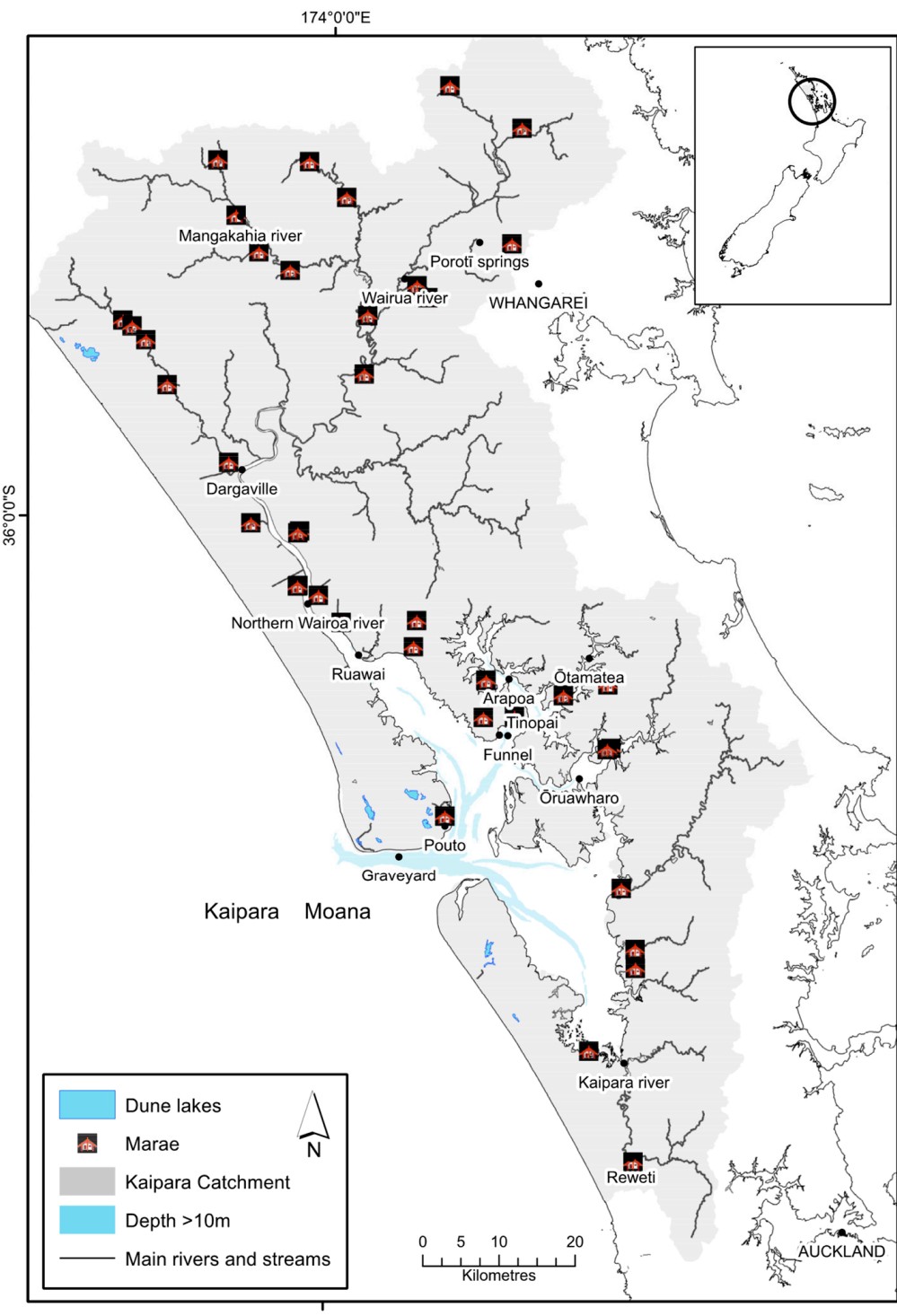

**Figure 1.** Kaipara moana and catchment (Inset. Aotearoa New Zealand).

## A brief pre-colonial history:
## Ngāti Whātua o Kaipara

**925AD** When the first fleet of migratory waka (canoes) travelled from the islands of the Pacific to Aotearoa New Zealand in 925AD, the men (including Te Tino o Maruiwi, Kupe and Toi te Huatahi) already found the Kaipara area settled by the first known ancestors of Ngāti Whātua o Kaipara (Tumutumuwhenua and Kui). 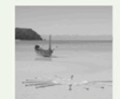

**1300** Māhuhu kei te rangi waka arrived in the Kaipara. Rangatira (chief) Rongomai and many of his crew settled area of land named Taporapora (inside the harbour entrance). Their descendants formed the nucleus of Ngāti Whātua. The original tribal name stems from a great grand-daughter of Rongomai, Te Whātua-kaimārie. 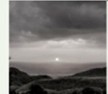

**1450** Another iwi (Ngāti Awa) arrived in Kaipara and settled in the area west of the Kaipara River. After 150 years of harmonious relationships between Ngāti Awa and Ngāti Whātua disputes over rangatiratanga (chiefly authority) lead to war. Through the conflict and intermarriages a new iwi – Kawerau – was formed. Sporadic inter-tribal warfare between iwi occurred throughout next two centuries. 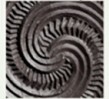

**1818** Another iwi (Ngāpuhi) under the leadership of Hongi Hika (equipped with muskets he gained through trading the European traders and missionaries) began to attack Ngāti Whātua people along the coastline. Ngāti Whātua were forced to retreat inland to escape the muskets, and only returned to their rohe following the death of Hongi Hika in 1828. 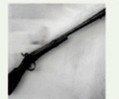

**1820** British missionary Reverend Samual Marsden travelled to the Kaipara and meet with prominent rangatira. However, there was limited contact between Europeans (Pākehā) and Māori in the Kaipara before the signing of the Treaty of Waitangi (Tiriti o Waitangi) in 1840. 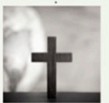

**1839** British colonial officials were aware that the demographic and military dominance of Māori that Britain would not be able to create and govern a new colony without the support and consent of rangatira; nor could Britain secure land for settlement without the assistance of Māori. The British Government instructed its official representative (Captain Hobson) to ensure Māori were protected from unscrupulous land purchasing practices by Pākehā. 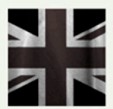

**1840** The Tiriti o Waitangi was signed by Captain Hobson, various English residents, and more than 500 Māori rangatira. This included rangatira from Ngāti Whātua. Most rangatira signed the Māori language version of the Tiriti. There were large differences between the English and Māori versions of the Tiriti. The English language version contained explicit mention that Māori were giving up their sovereignty to Britain, the Māori version did not. Instead, the Māori version concentrated on the principle of partnership between Māori iwi and the British Crown. Rangatira signed the Tiriti in the belief that their sovereignty rights (mana and rangatiratanga) remained undisrupted, and they were gaining the British Crown's protection from unscrupulous Pākehā as well as consenting to British residents living in Aotearoa. Both versions of the Tiriti included an article which guaranteed Māori would retain possession of their land, water, and other taonga (treasures); provision that Māori (if they wanted to) would sell their land directly to the Crown; and that Māori would be afforded all the rights and protections given to British citizens. As a result of the Tiriti Britain declared Aotearoa New Zealand a separate colony of the British Empire in 1841. 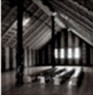

## Ngāti Whātua o Kaipara:
## 1840-2000

**1841** Rangatira of Ngāti Whātua believed that they (like other Māori leaders) were equal partners with the British Crown, and therefore they would legally recognised and fully involved and consulted in all decision-making, especially with regard to their whenua (land). The Crown assured Māori that it would provide iwi with protection, hospitals, schools, roads and other infrastructure in return for selling their lands. The Crown did not honour its promises to Māori around Aotearoa including in Kaipara. 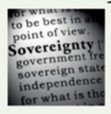

**1848** From 1848 to 1868, the Crown undertook a large land purchasing programme in south Kaipara. While Crown officials declared that low prices they were offering for Māori land would be off-set by benefits to Māori (provision of roads, hospitals and schools) these were either slow to arrive or never provided to Ngāti Whātau. Earlier, between 1841-1847 Ngāti Whātua sold land to individual settlers with the strategic aim of promoting development within their rohe, but continued to exercise their rangatiratanga over their lands and waters. 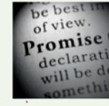

**1864** In 1864 the Native Land Court began hearings in south Kaipara. The purpose of the court was to force Māori to convert their land from communally-held (hapū) land tenure system to a British-style individualised land titles. The awarding of land titles to individuals (rather than hapū) meant it was easier for land to be sub-divided and sold. In addition, land surveys and court costs were very expensive, and Ngāti Whātua and other iwi were forced to sell land to pay their court-related debts. 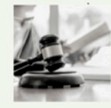

**1867** Ngāti Whātua constantly sought equal partnership with the Crown. The Crown established four parliamentary seats reserved for Māori in 1867. However, this did not meet the expectations of Ngāti Whātua and they continued to protest that the Crown was failing to honour the guarantees of the Treaty. During this time, European settlements grew in the Kaipara area and large-scale clearance of native vegetation and drainage works were underway. 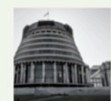

**1880** Ngāti Whātua sought to promote European settlement in the south Kaipara with the hope that it would foster development opportunities for its iwi members. Land was made available to the Crown for public purposes, including land to establish Te Awaroa (Helensville) and the railway line. Lands gifted came with conditions. The Crown promised to provide facilities (e.g. schools, roads) to Ngāti Whātua and return the lands when no longer needed; once again it failed to honour its promises. 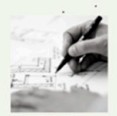

**1900** By the start of the twentieth century Ngāti Whātua retained ownership of only ten percent of their south Kaipara lands. The majority of their remaining lands were marginal hilly or sandy and unsuitable for horticulture or farming. Coastal lands at Puketapu and elsewhere were also compulsorily acquired by the Crown for coastal reclamation purposes. The last remaining large land area in South Kaipara held by Ngāti Whātua were also leased out (with its owners consent) for more than 50 years by a government appointed board. 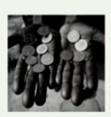

**1940** Ngāti Whātua were virtually landless by the 1940s. Due to poor living conditions, lack of educational and health provisions, and restricted economic opportunities, and racially discriminatory policies, most Ngāti Whātua lived in poverty. Many Ngāti Whātua migrated to Tāmaki Makarau (Auckland) and other urban centres in search of jobs. 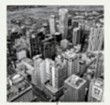

**2000** Throughout the 19th and 20th centuries Ngāti Whātua witnessed their lost lands being radically changed by Europeans. Their forests were felled, wetlands drained, and rivers dammed to facilitate the establishment of British-style agriculture and horticulture operations. The ecological transformation was fast-paced and pervasiveness; by 2000 less that 7% of native tree cover and less than 1% of wetlands remained. The Crown's colonial rule, based on its breaches of the Treaty, meant Ngāti Whātua were unable to maintain their ways of governing and managing their taiao (environment), based on their tikanga (laws) and mātauranga (knowledge). They were unable to meaningfully enact kaitiakitanga (environmental guardianship) practices and the health and wellbeing of Ngāti Whātua and its more-than-human kin declined as a consequence. 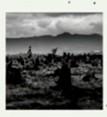

(**a**)

**Figure 2.** *Cont.*

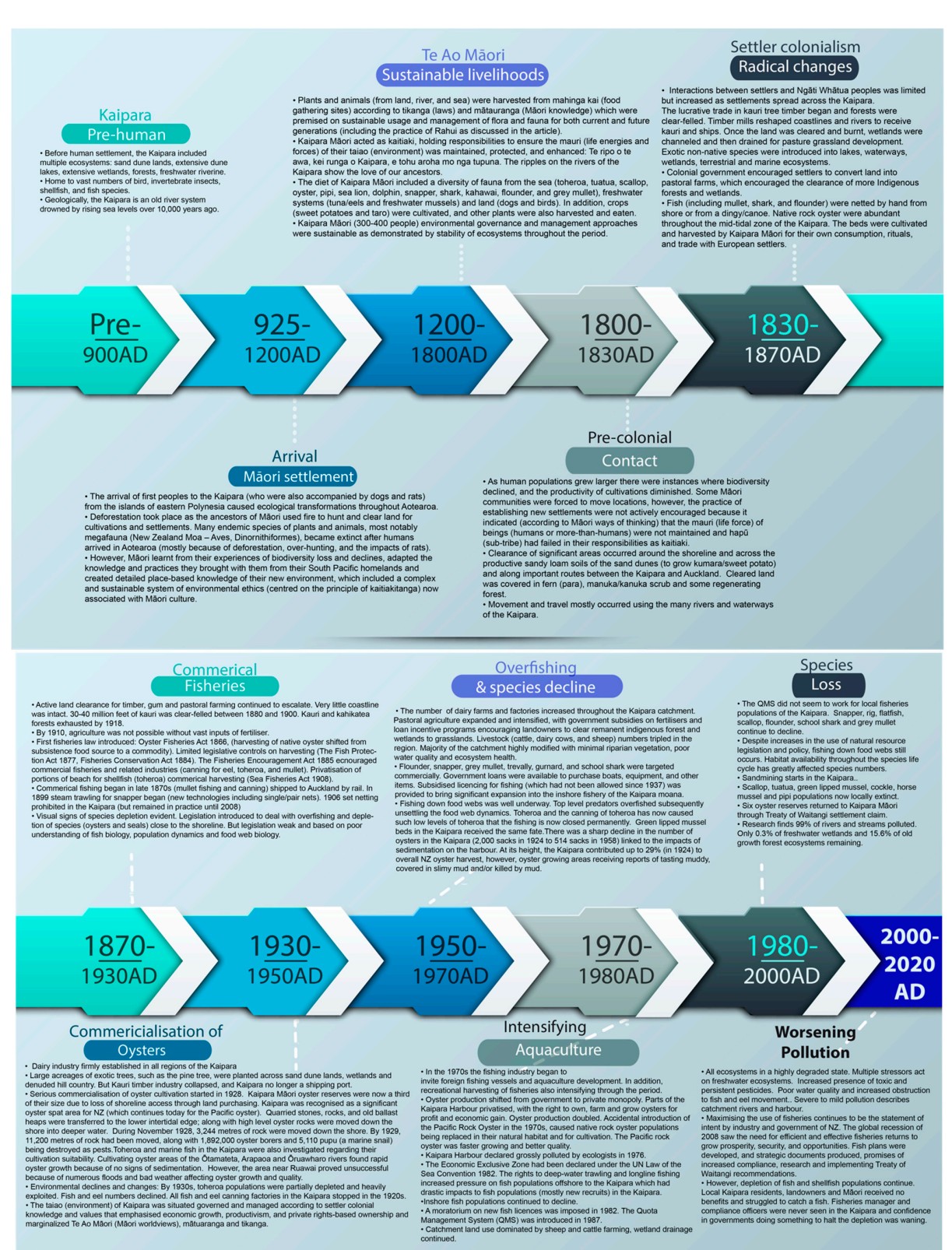

**Figure 2. (a)** Pre-colonial history timeline of Ngāti Whātua o Kaipara (Source: [33–35]. **(b)** Kaipara moana environmental history.

In te ao Māori, all beings are linked through whakapapa back to the beginning of time; this includes the gods starting with the primaeval parents: Papatūānuku (Earth Mother) and Ranginui (Sky Father) [36]. Whakapapa ties all entities together, both humans and more-than-humans, through kinship. The concept of whakapapa goes beyond simplistic notions of genealogy as being about biological descent (as per dominant Western ontologies and scientific knowledge) and instead the basis on which Māori socio-cultural, economic, and political systems of organisation [37,38]. Te ao Māori positions Māori as part of and sharing kinship with humans and more-than-humans (including plants, animals, soils, rocks, waters, seas, and supernatural beings), all of which contain mauri. Accordingly, humans are part of (rather than separate from) the natural world and possess specific responsibilities to care for their taiao (including all their non-human kin). Numerous Māori whakataukī illustrate the kin-centric, holistic, and relational worldviews of Māori wherein the mauri and wairua of humans and taiao are bound together [39–41]. A whakataukī used by Ngāti Whātua kaumatua, 'Toi tū te whenua, whatungarongaro te tangata the land endures, while the people come and go' illustrates two things: firstly, Māori cultural identity (as individuals and members of whānau, hapū and iwi groupings), as well as their mana, wairua, and mauri, are all inexhaustibly connected to their moana, whenua, awa, and maunga; secondly, through whakapapa, Māori are duty bound to care for their more-than-human kin just like they would for their human relatives. Māori obligations and responsibilities to care for their taiao is known as kaitiakitanga.

Kaitiakitanga is based on tikanga (laws), kawa (protocols) and mātauranga (Figure 3), all of which are embedded within te ao Māori, and provides the framework for environmental governance and management approaches [42]. Kaitiaki are individuals responsible for taking actions to protect or restore the mauri and wairua of their taiao and ensure its health (hauora). Kaitiaki practices are based on the premise that humans should treat their more-than-humans kin with the same aroha and manaakitanga as they would their human whānau [4]. For those of us (Vicky, Alyssce, Mina, Mikaera, Chris and Millan) who are the kaitiaki of the Kaipara moana, our duties and practices to care for the moana (and other more-than-humans) derive from our whakapapa, governed by our tikanga Ngāti Whātua (laws of our iwi), and directed at ensuring that the mauri of all beings is not only maintained now but also for future generations [43–45]. Whereas scientists operating from a Euro-Western ontological perspective describe environmental degradation (such as water or air pollution, and the impacts of climate change) as contributing to the loss of ecosystem function, scholars writing from a te ao Māori perspective position degradation as diminishing the mauri of rivers, seas, estuaries, mountains, forests, plants and animals (including humans) [46]. In the moana and awa around Aotearoa the notable declines in water quality, changes to water flows, increased sedimentation and plastic pollution, more frequent toxic algae blooms, spread of new invasive species and diminished biodiversity, are viewed as evidence of the diminishment of mauri. Moana is in a state of mate.

As settler colonial (or Crown) rule quickly expanded throughout Aotearoa in the mid-to-late nineteenth century, its (Western) ways of knowing, being, and doing came to dominate. Te ao Pākehā (the world or worldviews of Pākehā) was premised on binaries that separated nature from culture (humans) and positioned humans (specifically Pākehā) as being superior to and in charge of nature. Accordingly, te ao Pākehā fundamentally differed from te ao Māori, and the imposition of settler colonial rule meant that Ngāti Whātua not only experienced the loss of their lands, the trauma of watching their more-than-human kin be hurt (logged, burned, drained, and polluted) and lost (whole ecosystems destroyed and animals being made extinct) but also their mātauranga and tikanga being excluded [33,47]. The natural world for Kaipara Māori was (is) centred on reciprocal relationships with their more-than-human kin, but under settler colonial rule their more-than-human relatives became simply 'natural resources' to be commodified (capable of being owned, that was something unimaginable within te ao Māori) and exploited by humans. Thus, flounder and scallops became stocks. Forests became timber. Whenua became property. Whānau became individuals were owned (or not) property.

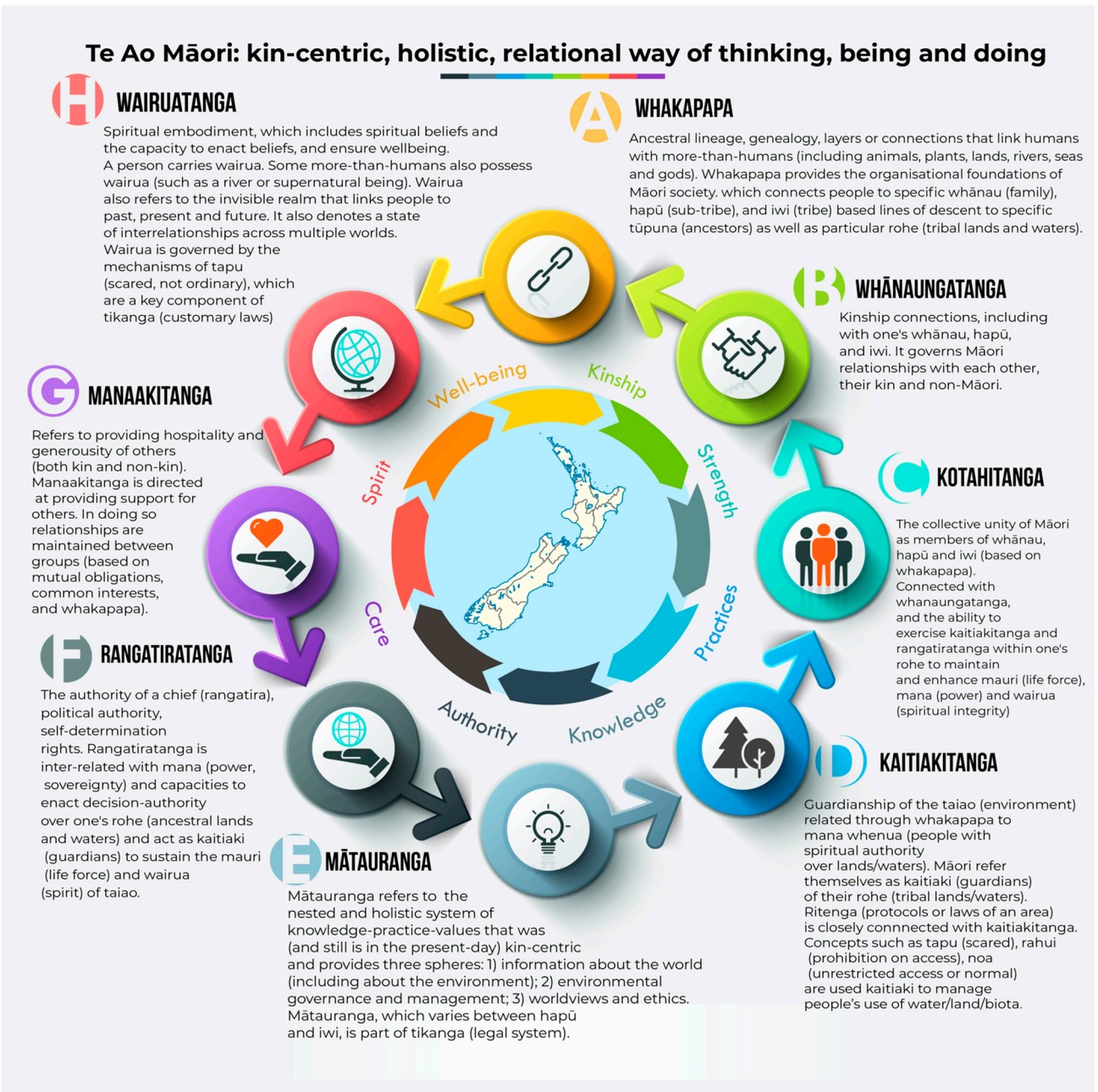

**Figure 3.** A schematic of te Ao Māori, the Māori worldview and ontology.

The clearance of forests involved either felling trees (if they were deemed economically valuable) or simply setting fire to them (if the tree species was declared low-value). Wetlands were drained with water and biota, as well as Māori cultivations and wāhi tapu (sacred sites) all removed in the name of settler colonial progress. Both former wetlands and forests were seeded with exotic grasses, fenced, and filled with exotic animals (cows and sheep). Although Kaipara Māori made various attempts to be involved in and benefit from the settler colonial economy, they not only encountered racial discrimination from individual settlers as well as government institutions but also witnessed how settler colonial actions were fundamentally in conflict with their values, worldviews and tikanga. Ngāti Whātua, like other Māori, spoke out about environmental changes they were witnessing and drew attention to the ecocide being committed, but settler-colonial authorities and individuals ignored or marginalised their voices. Instead, in the twentieth century the Kaipara ecological transformations further intensified as the native timber industry

declined, transportation linkages to Auckland city improved, and the dairy industry expanded. Industrial agriculture became the norm characterised by more intensified farming methods (including large herd sizes and the heavy use of fertilisers, agri-chemicals, and supplementary feed). For instance, herd sizes have markedly increased in the twentieth and into the twenty-first centuries, with the average herd size for a Kaipara farm growing from an average of 37 cows in 1920 to 400 cows in 2017 [48,49]. Likewise, fertiliser usage is growing more and more widespread throughout the Kaipara area (a national trend), which is also linked to declining water quality (both in rivers and moana). As the fertiliser breaks down into its nutrient parts (nitrate, nitrite, ammonia(um), phosphate, phosphorous and heavy metals such as copper and lead), these elements are picked up, from residing in the soils and flow off the land and into the water. Between 1991 and 2019, estimates from sale data of nitrogen based fertilisers applied to land have increased by 629 per cent [50,51].

It is unequivocal that land-based activities caused and continue to result in the significant environmental degradation of the moana. In addition to industrial agriculture and horticulture operations in the area, urban areas (Helensville and Dargaville) with their mix of residential and commercial premises contribute to run-off into rivers, streams, and coastal ecosystems; run-off including sediments, nutrients (nitrogen, phosphorus) and pollutants (E.coli bacteria, heavy metals, dioxins) from the land combines with wastewater (including treated sewage) that is deliberately pumped into waterways (adding further pollutants). As early as 1976, scientists were labelling Kaipara moana as a heavily polluted environment [52]. In 2022, its ecosystems, habitats and species remain in a state of ongoing ecological crises, being physically and structurally altered and destroyed by settler colonial-led activities for more than a century. The wairua and mauri that resides within and between Kaipara's terrestrial, freshwater and coastal ecosystems, and the Māori people of the Kaipara are so badly disrupted, fragmented and degraded, that some mana whenua are uncertain if the wairua and mauri can ever be restored.

## 4. Decolonising Methodology: "Thinking with Kaipara"

Five Māori researchers' narratives (three wahine/women (Vicky, Alyssce and Mina); two tane/men (Millan and Mikaera) are presented. These stories include personal experiences, whānau/hapū/iwi histories and whakapapa, and were created using the decolonising methodological approach known as "Thinking with Kaipara". The decolonial "Thinking with Kaipara" strategy, underpinned by Kaupapa Māori, eco-feminist and antiracist theorising, was created through a collaborative research project that involved both Pākehā (the lead author, Leane) and Māori researchers. The research pursued relational, embodied, and transformative methodological practices to know and interact with a specific place (Kaipara moana) through situated knowledges [53,54]. "Thinking with Kaipara" focuses explicitly on acknowledging te ao Māori and mātauranga, multiple subjectivities, and human and more-than-human relations all of which have been marginalised by settler colonialism and neoliberal capitalism.

This research positions Māori as knowledge-holders (rather than objects of knowledge) and actors, along with Kaipara moana, who possess mana and are part of a holistic and interdependent system [55,56]. Our writings and representations of environmental changes within the Kaipara moana, as well as its catchment and ecosystems, take into account beings that are living and non-living. We employed diverse research methods and materials to document and narrate environmental degradation and changes as well as our efforts to restore the health and mauri of the Kaipara.

Within this framework, the physical and metaphysical realities are inextricably linked to people's experiences and interactions with places and environmental changes [42,53,57,58]; as Haraway [59] argues, all matter (inanimate and animate) is important as it forms part of the cosmos. Consequently, our research project explicitly recognises and (re)presents a socio-cultural, political, spiritual, and environmental ontological-epistemological approach that disrupts and challenges the orthodoxy within academia (research design, methods, and findings). Our work adds to the emerging body of scholarship that promotes Indigenous

storytelling and story-work practices. Such activities are most frequently the means for Indigenous peoples to retain their socio-cultural identities, as well as their links to their families, ancestors, tribal groupings, and local land-/water-/sea-scapes [60–63]. We (as researchers and storytellers) critique and oppose settler colonial violence (against human and more-than-human beings) and decolonise thinking, being and doing by employing methodological methods derived from Indigenous ethics, values, and knowledge [62,64–66].

We (the researchers) have all witnessed the violence of pollution inflicted on Kaipara ecosystems. We were (and continue to be) active in ecological restoration projects that rely heavily on connections with wairua and mauri of the taiao and the human and more-than-human beings who live with(in) it. Data collection (story-work) was conducted between 2017 and 2019 and involved informal "conversations" (rather than more conventional types of research data collection techniques such as surveys or interviews). We talked, sang, drew, texted, photographed, emailed, wrote, walked within and outside the geographical boundaries of the Kaipara catchment, and produced a diversity of socio-cultural artefacts (the empirical data of this paper). The primary sources (transcriptions, text messages, genealogies, field notes, field observations, lyrics of songs, pictures, drawings, and photographs) were supplemented by secondary sources (reports, social media, and social statistics). While the focus of our storytelling process began with discussing the pollution of the Kaipara moana and all its kin (which includes flora and fauna, animate and inanimate matter, physical and metaphysical), it quickly extended to include the multiple forms of violence (ecological, colonial, inter-personal) that we witnessed/experienced.

The lead author (Leane) started individually with co-researchers and began a dialogue, primarily with kanohi ki te kanohi (face to face) hui, and then continued through email, text, phone, Skype/Zoom calls, and Messenger. Dialogues involved discussing lived experiences. Such experiences included their memories, histories, emotions, spirituality, and aspirations for the future. Through an iterative and non-linear creative process, we talked, considered, debated, revised, and finalised the narratives (outlined later in this paper). Some narratives are presented in first-person (as one person or two people) and others in the third person. We sought to (re)present experiences, stories, and things without representation [67] and go beyond dichotomised language and thinking, and achieve a feeling, attitude, and sense of openness [68]. The linguistic and visual registers were altered (sometimes from first person to third person) to make it easier for the readers to trace the narratives. In this research, we felt, touched, listened, and utilised all our senses to detect ecosystem changes, the presence (or absence) of more-than-humans, and the entanglement of ecological violence with socio-cultural, and embodiment of pollution. We attempted to write in a way that allowed the data to speak for themselves. In the next part, we offer our stories.

## 5. (Un)Heard Voices

In this section, we present a set of stories composed by the five Māori researchers involved in this research.

### 5.1. Story: Why One Scallop

A short story by Mina Henare. Tinopai. Kaipara Moana.

A young child asked me why is there only ONE scallop in Tinopai. To this, I replied, well,

ONE man bought land with a wetland in it and bulldozed the wetland, so the man next to him did the same and destroyed the other side of the wetland; Wetlands protect our Kaipara Moana.

ONE man has a pine forestry whose contractors felled their pines into our waterways, their machinery leached oil, their silt traps failed, and the tannins that came from the forestry went directly into our Kaipara Moana.

ONE man caught 1000 scallops in one day when the scallops were there only three years ago.

ONE farm directly above the scallop grounds sprayed poisons to kill the weeds and then sprayed his farm with nitrates and phosphates, all of which were washed into the Kaipara Moana when it rained the next day.

ONE farmer, all the way up the estuary, clear-felled his native trees down to the waterway, and you could follow the debris and mud trail all the way to the Kaipara Moana.

ONE mangrove forest was destroyed so that the beach would look better; mangroves prevent silt from entering our Kaipara Moana and are nurseries for fish.

ONE Council is millions of dollars in debt and does not have enough money to cope with the largeness of the Kaipara Moana.

ONE Regional Councillor is sitting in his office a hundred miles away. He is qualified in paperwork and argument but not in our Kaipara Moana.

ONE Kaipara Moana is so large that it has TWO Regional Councils that think differently.

ONE scientist found one scallop when he searched the whole of the Kaipara Moana, and that scallop was dying.

And when you put all of these men together and multiply them by the area of the Kaipara Moana—that is why there is only ONE SCALLOP IN TINOPAI.

Now that ONE SCALLOP can join our 'no more' list for Tinopai: No more mussels; no more scallops; together with our 'barely there' list: flounder, stingray, dolphins, orcas.

*5.2. Story: Tuna Saved My Life*

Tena Koe.

Ko Whatitiri te maunga. E tu nei I te āo I te pō. Ko Waipao e Wairua te awa I rukuhia, I inumia e ōku mātua tupuna. Ko Maungarongo te marae. Hei tangi kit e hunga mate. Hei mihi kit e hunga ora. Ko Te Uriroroi. Ko Te Parawhau. Ko Te Māhurehure ki Whatitiri ngā hapū. Ko Ngāpuhi-Nui-Tonu te iwi. Ko Millan Ruka ahau.

You know your backyard when you haven't moved out of it too often. Although I was born in Auckland, the Wairua and Mangakahia rivers are my turangawaewae.

Thirteen of my father's brothers and sisters come from here, Porotī.

Born on the river . . .

A lot of them are buried . . . on the side of the river including my grandfather . . .

So that's my place.

Millan's whakapapa relationships with humans and more-than-humans (sky, soil, river, land, gods and biota) are incorporated into Millan's mihimihi. Millan is watching, listening and imagining. He is standing at the source of te awa o Mangakahia one of the largest rivers that flow over 80 kilometres to the Kaipara moana. "I can look at the water all day like this." He is speaking about his genealogical home; the places he was born and raised. Millan wonders where the water comes from and ponders on the fact that it keeps coming. "[I want] to walk right up in [to the awa] and just see where it comes from," he says. Millan knows where it goes. This place derives mātauranga, tribal wisdom and knowledge, that enlightens different aspects of the world around Millan. Through a process, Millan has taken to learn, understand and know and practice this mātauranga.

I can remember the days of the Wairoa and the Mangakahia rivers being crystal clear, and swimming in, and drinking from the river. You never thought twice to have a drink from the river . . . and streams. We'd sit on the bank of the river at home and just look down and spot an eel if you sat there long enough. Well, you couldn't see anything. It's . . . like a green soup . . . . . . . They are nitrate laden. E.coli

[bacteria] laden. Way past the drinkable stage, so taking your health into your own hands. Just seeing the depletion and the non-sustainability of the rivers,

just in my time . . . .

Just a few metres on the other side of the bank. [Whole herd of cows] having a mimi everywhere. Excrement. Because they do that when they're curious and just standing around, may as well have a crap while I'm here.

I started to cry.

Realising the enormity of it. This is unbelievable, knowing my rivers and even that place as a kid, and hard places to get to . . .

you won't . . . **see** . . . the Wairua river unless you take a side road. So, you

don't even . . . **see** . . . these rivers—the Wairoa, the northern Wairoa,

feeding into the Kaipara—you won't . . . **see** . . . that until you get down to

Dargaville. So, you don't . . . **see** . . . these rivers. You don't . . . **see** . . .

what's happening to them. Unless you're . . . **on** . . . them.

The switch just went on . . . I'm going to report on . . . our **rivers**.

I couldn't believe how much it had changed in a very short time because [of] the impact of dairying. Fonterra had evolved from being just a rural dairy companies to amalgamate into the huge giant they are now. Even in the [ten years] I was away, I saw many, many farms where it was one farmer just on the 220 acres or so, to being, that farmer had gone, or his son had taken over, and it amalgamated with the farm next door, left and right, to make it a bigger farm with 500 cows and 600 acres sort of thing. Intensification started there because they had big mortgages, but to me, they were relying on the capital gain as much as gain from the product. The intensification was huge. It was overnight.

Millan thinks about the (slow and invisible) violent journey of disruption, degradation and manipulation the wai takes before it reaches the Kaipara moana. He explains te mana o te wai is denigrated, usurped, disconnected, and extracted. He describes his responsibility and obligation to protect the mauri of the Mangakahia and Wairua for current and future generations [69–72]. Te mana o te wai has not been protected or ensured, so Millan reports on the status of the awa, a responsibility as kaitiaki of the rivers. He creates GPS maps and photographs cattle stock movements, stock defecation and urination along the edges and in the awa. He submits the reports to the Council 0800 POLLUTION HOTLINE. His first report was in September 2011 on the Wairua power station canal. Sometimes reports include up to 140 photos that have GPS information attached. The report does not identify (landowner) names. It is not improving—the awa or the government response. Millan cannot drink and eat safely from the awa like he used to do as a young man. The physical devastation and degradation of his awa runs deep for Millan knowing that as a young man he could access and see the awa, drink from the awa, swim and get a feed of tuna from the awa; and financially support himself and his family. "Tuna saved my life," Millan says. First, as a boy when his father returned, a WWII prisoner of war veteran and then as a man during the 1987–88 financial depression when no one needed a builder, just food. Evidence highlights that Indigenous peoples experience higher rates of personal trauma than Pākehā/Europeans/Whites and suffer a higher incidence of lifetime trauma [57]. The harm and grief Millan experiences from settler-colonialism is a daily embodiment, a characteristic of colonial ecological violence [12].

Millan's day job is reporting on the state of the Kaipara rivers, particularly the Mangakahia and Wairua, in the northeastern parts of the catchment (Figure 1). Millan's night job involves writing reports, emails, and submissions on resource consent applications, cultural evaluation reports, going to hui and hosting hui. This mahi is a responsibility to care for the awa and wai. "Easier to keep all Māori out," he says, referring to the Resource Management Act (RMA), Aotearoa's environmental development legislation, and the "crawl process"

he, his whānau and hapū have to go through to protect their most sacred region, Porotī Springs [73]. This fight has been ongoing since 1959 when nonMāori wanted to take water from the spring. To keep Porotī Springs spiritually intact and maintained by the hapū is a battle that involves Millan. It is a responsibility. A calling.

> "We see ourselves after much study being the sentinels on the river for Ngāpuhi. One of my cousins pointed out we've always been a fighting line, Te Mahurehure. Where there are other hapū [that are] growers eh, harvesters of kaimoana or growers of food eh, gardens, but not us. Predominantly we're just fighters."

Millan utilises this fighting metaphor to describe his engagement with the RMA and the local governments that administer the Act. The "trenches of the RMA", he says. The many volunteer hours mentally, physically and spiritually involved in his approach make it more of an intellectual-spiritual journey rather than a collection of facts or an 'archive of information' [36]. Moving between the Māori world of knowing, being, doing, and the Pākehā ways of knowing and doing as represented by the RMA, these are different worlds, "not one world viewed differently" [74], and as he says to me on the phone a "crawl process". The inequalities and cruelty of the RMA run deep for Millan, Porotī Springs, and whānau. He and members of his tribe use multi-level approaches [75] to Titiri claims-making, prosecutions and disputes, spanning local, regional and national initiatives that balance negotiation, strategic partnerships, civil action and presence on the awa. Millan gets many requests a day to speak about his mahi, the awa, pollution, and his river patrols. He moves between meetings on developing freshwater management policy at a national level, to writing reports about the pollution photographed in the river and bolstering strategic partnerships with research agencies. All while maximising his cultural and spiritual objectives. (see Supplementary Materials for the whole story).

*5.3. Story: We Survived off the Kaipara Moana*

Vicky introduces herself and delivers a karakia before we get started. Karakia, a prayer, invites more-than-human forces to enable conversations to proceed well and with gratitude; and to enfold forces (bodies, spaces, ancestors, earth, and sky) into productive engagement [68]. Relations with and between all things, including human beings, is a Māori worldview [74]. Vicky, the human, begins.

> Ko wai toa te maunga Tokatoka. Ko wai to ate awa Wairoa. Ko wai toa te waka Mahuhukiterangi. Ko wai toa te iwi Ngāti Whātua. Ko wai toa te hapū Te Uri o Hau. Ko wai toa te marae Waiotea. Ko wai to ate moana Kaipara. Rangimarie Harris nee Connelly te mama, Guy Harris te papa. Glen Miru te tane.
>
> Ko Vicky Miru ahau.
>
> I'm passionate about the environment, and I care about the environment and worry about the environment, especially with my culture as Māori and when we gather kaimoana and things like that. I've noticed a lot of concerns about the moana and what's been happening in the moana. That's how I got involved. I got involved through Te Uri O Hau and Mikaera and a whole lot of other people that started this journey with them. We all had the same views and concerns and things to do with the moana and the environment. I just got involved because I care about it.

Vicky talks about her life on the Kaipara moana and her kaitiakitanga responsibilities, which encapsulate an ethic for caring for Kaipara nature as well as all other more-than-humans who reside within the moana. Over the centuries and decades, the Kaipara, Vicky observes, provided Māori with so much kaimoana. However, the amount that can be harvested is becoming less and less. Vicky watched ongoing basis since first coming to Tinopai, to raise her children, over 30 years ago.

> "What did you value about it?" Leane asks.
>
> "There was plenty of kai," Vicky responds.

**AND**,

"But it's just the parū in the moana. That's what got [me] into this kaitiaki journey. Protecting our kai, protecting the animals and the water and the fish and things like that. That's how I got into it; I think." She responds.

**AND**

"Just the feeling of having to fight for our rights to keep the place as it is and not—I don't know how to explain it. But it's just in me that—that's something I wanted to do. I just wanted my kids to have a clean life sort of thing with clean food and good food and things like that and just live a decent lifestyle. The lifestyle we had in Wellsford wasn't doing us any good, so I came up here."

**AND**

"[S]ediment, that's runoff from the land, eh. That builds up and builds up and covers our shellfish and kills our shellfish . . . . All of a sudden, it's there. It's everywhere now it's all mud and silt and siltation, and it's just not—it's not nice. . . . You go out there, and it's just not nice. You get all this yucky stuff out there. It's not nice. That's another indicator. I suppose you could say."

**AND**

"Quite a few warning bells were going, that this isn't right," Leane says.

"Yes. The snapper, for instance, you had to go out past the bar to catch big twenty-pound snapper. In here [Tinopai] all we caught was three or five-pound snapper or smaller than that. I thought there must be something going wrong," Vicky responds.

**AND**

"That's what they were taking, the mullet and the flounder. Not so much the snappers. One time we had no mussels. I don't know why, but then they came back again. We used to go down to the beach and get a kai of oysters or something. It used to be all papa-rock. Now or even ten years after that, now it's all mud and silt and siltation and just—it's just not—it's not nice," she says.

**AND**

"So, you, the whānau, put the rāhui down?" Leane asks.

"Yes," Vicky responds.

**AND**

"That was quite significant, wasn't it?" Leane says.

"It was. Yeah. It worked." Vicky responds.

**AND**

"Because that was a traditional Te Uri O Hau, Ngāti Whātua rāhui. It wasn't a fisheries Pākeha law?" Leane asks.

"No," Vicky responds.

**AND**

"The next rāhui was [20]'08 or [20]'09 for the turbine project?" Leane says.

"Rahui out here [Tinopai], that we had out here worked. The one up there [Pouto] worked as well because they haven't gone ahead with any turbines after that after we did the rāhui up there. I believe in the rāhui because it does work. Well, it's worked for us anyway, so far so good." Vicky states.

**AND**

"It's good management for the Kaipara. It will be good management. It's a better way of managing things than the way they are doing it now. To me anyway, how I—that's how I think, anyway. It's a way of managing our stocks, our fish stocks, our kai moana. It worked in the old days." She responds.

**AND**

"I was there with the waiata. Also, I'm connected to the Kena whānau in Pouto." She responds.

**AND**

"If they put on a turbine, that's going to affect the fish as well and the dolphins and the orcas that come into this moana. That's why I went up there to protest." She responds.

**AND**

"Yeah. It was awesome. We all stood around, held hands and the kaumatua did his prayer and put the rāhui down. We all stood around, and it was awesome. Had all our banners and things like that. It was awesome. It was just amazing really. I was really rapt by what happened up there. It was awesome what they did up there. And that's another thing, community. It's a community. It's good in this town because it's a good community; here [in Tinopai] as well."

*5.4. Story: Section Five [of the RMA]: "This Is Bullshit"*

Mikaera has put his body on the line for Kaipara moana. Police have charged Mikaera with inappropriate use of a firearm. That needed to be done, he says. He has been called crazy, narcissistic, angry, welcoming and caring, as well as, confrontational, kaitiaki and kaimahi. Mikaera feels oppressed, powerless and angry. He takes peace from the guidance of his tupuna Haumoewarangi and Rangiwhapapa both leaders of Ngāti Whātua. Mikaera ponders his priorities. "It's on my bucket list", Mikaera says regarding kaitiakitanga at the local scale. "That is my focus. Something that I've been doing for decades anyway, but when you've had an experience that I've just come out of, you think of what your priorities in your life are. What do you want to do before you no longer exist?" He is in the middle of radiotherapy for lymph node cancer. "It's huge, and you need to think, what is important to me? What is important to me?" he says. Mikaera resides on whenua that has never been sold to Pākēha settlers. This is his place, which he calls home.

"Look . . . I've done this shit for years . . . I've done it for decades", he says regarding the calling out of the environmental destruction occurring right in front of his eyes in Tinopai (Figure 1) and the Kaipara moana. That "shit" is watching "the environmental policeman" who are "out to lunch" when it comes to upholding the RMA (Resource Management Act) and enabling hapū to participate in the RMA consenting and development decisions. The Act recognises Māori environmental values in the purpose of the Act [76]. However, the RMA juridical framework constraints Māori ability to exercise rangatiratanga primarily because the Act does not define Māori property rights (customary tenure) regarding water, coasts, air, land and soil. Māori are not elected representatives on local governments responsible for Kaipara moana management decisions. Rural land interests established for agricultural exports continue to exercise strong dominance in local government decisions. To promote their values and aspirations for the management of the natural world, mana whenua representatives like Mikaera are constrained to primarily advocacy work (Mikaera held the position of the Te Uri o Hau Settlement Trust representative for two years on the Northland Regional Council Te Taitokerau Māori Advisory Committee). For Mikaera, his particular challenge is the local government not doing their job of upholding the purpose of the RMA in sections 5, 6, 7 and 8. All these sections encompass mana whenua values such as the regard and provision for the relationship of mana whenua and their culture, traditions and ancestral land-seascapes, water, waahi tapu and taonga; and to have particular regard to kaitiakitanga (s7(a)) and safeguarding the life-supporting capacity

of air, water, soil and ecosystems (s5). Mikaera has had particular regard to the RMA and its "shit," he says, for a long time.

The "shit" that Mikaera and the voluntary marae-based resource management unit (RMU) he is involved with, is similar to what Millan does: responding to resource consent applications (for land development, industry establishment like underwater marine turbines, wastewater management, land use change); monitoring and reporting of consent conditions (for example, wetland protection, mangrove removal, riparian planting, wastewater and water quality monitoring); identification, protection and maintenance of areas of natural significance (for example, wetlands, waterways, estuaries, mahinga kai and wāhi tapu); creating and maintaining relationships with the Tinopai community; making local government accountable, and the integration and collaboration of local government organisations different plans and rules for resource use. Like Millan, Mikaera's and the RMU's efforts are unable to alter inequitable governing regimes even after governments have recognised the concept of kaitiakitanga in the RMA [77]. Recognition of difference may have been given in the RMA and fisheries legislation; however, scholars find that injustices and inequalities remain for many Indigenous peoples and cultures [78]. Mikaera is in pursuit of a type of transformative justice that reconciles ongoing commitment to dialogue, longterm relationship building and creation of a space for self-sovereignty for the resurgence of Indigenous Māori law, mātauranga and practices.

The first signs of local inequalities surrounding fisheries management in the Kaipara were when Kaipara fishing stalwart, George Pook, left for Australia, Mikaera says. They were a family born on the waters of the Kaipara. After 30 years of commercial fishing on the Kaipara, he believed there was no future for him in the Kaipara. This was in the early 1990s. The Quota Management System (QMS) was introduced in 1986 to address fish and shellfish declines and collapses. The QMS was to put the power into the fisher's hands to control the tonnage of fish they took/take by using quota (ownership right) and annual catch entitlements (access right). Since the first implementation of the Fisheries Act in 1908, Indigenous Māori rights have been negated, and the QMS was no different [79–82].

For Mikaera, "we were shafted again" and his experience of the QMS is recounted in Barry Barclay's feature-length documentary film, The Kaipara Affair [83]. The spatial conflict between Māori and (local and itinerant) commercial fishers came to breaking point: "They were raping and pillaging right in front of my marae", Mikaera says. He was honest about what happened next: "I put four shots into the air. [Because] about four 450 metre set nets were used in front of the marae. They left after that", he knew who they were. Local fishermen from up the river: "They came back again. I rowed out to them. I took my oar, but he used his assault rifle [on me] this time."

"[I] pull out a gun and fire a **bullet** and now everyone is listening,"

"This is a **bullshit** society."

Things were heating up on and off the water: "Would have been war. You go to war when the law doesn't mean jack shit," he says. Things were frightening, and unsafe, and people were angry: "How can a system allow 6,000 kilometres of set net." Such fishing methods were not indicators of sustainability. Discontented individuals released feelings of marginalisation, injustice and inequality. It was generally felt that the QMS was benefiting a few at the cost of the many [80], including the ancestral fishing grounds farmed by Mikaera's whānau for over 400 years. Indeed, this was considered a typical result for many Māori fishers after the QMS implementation [82]: "[We] were [are] starving right on the shore," says Mikaera.

Māori lore (tikanga Māori) was required, Mikaera says. The rāhui management tool has been a forgotten tool for Māori since 1887 when the first Crown law for fisheries was implemented, says Mikaera. Rahui is for conservation purposes where a mauri stone was used. If the rāhui were not followed or trespassed, there would be punishment by the Atua and utu on the whānau. In 1997, kaumatua placed a rāhui in the 'Funnel' (Figure 1), the area adjacent to Mikaera's marae, which included the Tinopai village coastline, across the

channel (or funnel) to the shoreline on the opposite side. It was a protective rāhui where no commercial fishing was allowed.

> The (Pāhēka) Law [Fisheries Act 1996] then assisted the community of Tinopai and Kaipara with a two-year closure.

Mikaera says, "in the end, we were shafted. It was very sad." The Crown made no improvements to the sustainability (biologically and socially) of Kaipara fisheries. The science was lacking and what science they had, did not improve the fish and shellfish (e.g., flounder, mullet, scallop, oysters) or their habitat. Joint and shared efforts by the local Pākeha and Māori of Tinopai and wider Kaipara were a force to be reckoned with because such an alliance had not been publicly seen before. A collective had been forged. However, whānau and community social values and living values were not respected,

> "We were starving right [here] on the shore." Mikaera continues,

> "This kōrero is so huge and large that you can not talk about it in one session."

> There is corruption at play; exclusion and racism he says.

The Waitangi Tribunal (forum to address historical injustices) freshwater reform analysis between 2003 and the present day [84–86] found that the RMA and its allocation regime are not consistent with and breach Treaty of Waitangi/Tiriti o Waitangi principles, including the principle of equity. Māori have been prejudiced by the ongoing failure to recognise their proprietary rights. Institutional and structural barriers have prevented participation in the first-in, first-served allocation system of the RMA, and the (Treaty) partnership in allocation decision-making. Economic opportunities have been foreclosed by the barriers to access water. The RMA makes a proviso for pre-existing rights of farmers but does not do the same for Māori, and does not otherwise recognise or provide for Māori rights of a proprietary nature [84].

> "[It's] another manifestation of institutional racism," says Mikaera.

> "We have to put up with this shit...every bloody day. They [Council/Crown] turn their cheek the other way. When they're been found to be wanting, they've created a problem. They just turn the other way. There's no response. Absolutely nothing. Every conversation that I'm having from now on . . . with people from the government, I'm shoving the law down their throats. Because really, what I'd like to say to them . . . if you're not prepared to uphold your legal obligations of the RMA well then get out of the bloody office."

(see Supplementary Materials for the whole story).

*5.5. Story: Te Aō Māori, Te Aō Pākehā—I've Been Brought Up in Both Worlds*

> Ka tangi te titi, ka tangi te kaka, ka tangi hoki ahau, tihei mauri ora. E tū ana au ki te taumata o Muarangi e tū takoto rā. Ka titiro whakararo iho ki te iwi o Ngāti Whatua. I te waka o Mahuhu ki te rangi. E rere nei I ngā wai karekare o Kaipara. Ka ū te waka ki uta, ki te marae nukunuku-ātea o Waikaretu. Kia rongo I te reo pōhiri o Te Uri o Hau. E mihi nei, e karanga nei. Ko Alyssce Te Huna ahau.

Above is my (Alyssce) Māori mother's whakapapa. My other whakapapa, which is not written here, is from my European Pākehā father. I was given a taonga by my mother. It was my aunty's (my mother's older sister's) taonga. My aunty was my godmother and my kōtiro, Olivia, carries her name. The taonga is two manaia intertwined (Figure 4). An ancient mythical being with a bird's head and a human form. It is the messenger between the earthly world of mortals and the domain of spirits. It symbolises the strong links between spirituality and the spiritual world. It is the holder of spiritual energy and the guardian against evil. The three fingers of the manaia represent life and death while the fourth finger represents the afterlife. The taonga represents whakapapa and spirituality. I feel it keeps me safe and centred and gives me the strength to work through the conflicts, trauma, and opportunities I face day to day, living in two different worlds. I do not know

my Pākehā whakapapa, it does not give me strength, it does not nourish me, however, being brought up in te ao Māori has given me the strength and courage to know who I am and help me navigate through two different worlds.

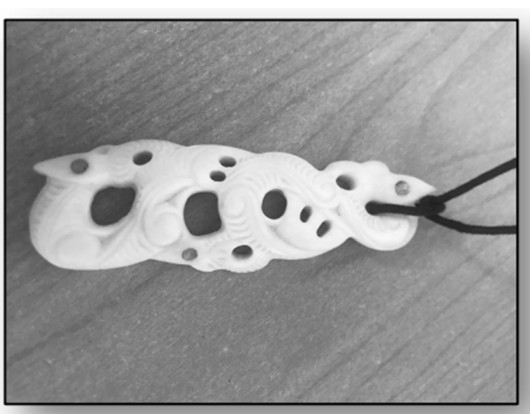

**Figure 4.** Alyssce's taonga.

I grew up predominantly in te ao Māori except when it came to my education. I was put into mainstream primary and secondary education because my grandmother wanted me to learn the "Pākehā ways". I was one of the only Māori doing science at high school. Science proved to be challenging but I loved it. I thought, this is so interesting, and I would always enter local school science fairs. I wanted to learn more about that world and work in that world. Many of my high school friends were going to university, so I thought, why not give it a try? I was one of the first in my whānau to achieve university entrance at 17 years old and graduate from University with a Bachelor of (te ao Pākehā) Science. For the first time at university, I experienced racism, I was called a plastic Māori; a Black bitch. It was uncomfortable.

What I came to realise is that working in the Western science world would mean letting go of my te ao Māori worldview of observation and interpretation; my mātauranga, language and tikanga. I was leaving behind subjectivity, spirituality and whakapapa to a world dominated by objectivity and the separation of nature and culture. I have experience working in both te ao Māori and te ao Pākehā centred organisations. My first job was as an environmental officer for my hapū, a post-settlement organisation, Te Uri o Hau Settlement Trust. I was so excited about the role as it was te taiao focussed; working alongside my whānau and marae to make a difference. However, after suffering massive burn-out mainly due to internal politics, I left to have my first child. I then took a job in te ao Pākehā, a directorate administrator role for the degree in horticulture and forestry at the education provider, NorthTec.

For better opportunities, we moved back to Palmerston North (where I did my tertiary studies) and continued my career in te ao Pākehā as a treaty ranger for the Department of Conservation (DOC). It was my first full time job since I had Olivia and I had the confidence that my knowledge and experience of both worlds would serve this role well. I wanted to use the resources we had in DOC to help empower and transform Māori. It was a new role for DOC in recognition of the new Treaty of Waitangi settlements in the region. Because there was no footprint I designed and shaped the role to support the uplifting of Māori (rather than oppressive and colonising). This was challenging for me and my managers to navigate. Iwi acknowledged me for the work and passion I had, however, DOC did not. The low pay and the lack of acknowledgement and support for the treaty partnership work I did were the main reasons I left this role.

In 2019, I transferred into a technical advisor role at DOC. I was excited because I could finally use my western science knowledge and zoology degree and apply this to the work. I wanted a break from being embedded in te ao Māori mahi and wanted people to recognise me for other skills. I coordinate and support 18 species recovery groups across Aotearoa NZ

ranging from amphibians, reptiles, and plants to many bird species. The groups provide te ao Pākehā technical and scientific advice for certain species/taxa recovery to internal DOC staff and other government and non-government organisations. Species recovery takes a collaborative approach in Aotearoa NZ recognising that DOC can not restore an endangered species alone. Thus, members of the group include a mix of DOC staff, whānau, hapū, iwi and external stakeholders such as Crown Research Institutes, community groups, and universities. Te aō Pākehā science is still the dominant way to describe the status of our environment, with limited mātauranga regarding our taonga species being acknowledged through the characterisation of a species/taxa.

When I reflect on my te ao Māori work spaces I felt nourished and accepted. However, I experienced injustice, racism and a sense of jealousy. When I reflect on my te ao Pākehā work spaces colonisation is rife, the workplaces are patriarchal, and people are institutionalised and compliant and complacent with working to bottom lines that have been informed by science and not the whakapapa and mātauranga of our taonga species, land and places.

> The colonisation of Māori saddens me.
>
> The jealousy, deceit, nepotism and standing,
>
> on each other, to get to the top must stop.
>
> The constant need to educate nonMāori staff
>
> in te ao Māori is tiring and a waste of my oxygen.
>
> It is never used in the way it should be.
>
> They pay 'other' experts hundreds of dollars but nothing for the mātauranga I hold.
>
> My wairua is not nourished, nor whole.
>
> I feel conflicted, unsafe, unheard, unsettled, lost,
>
> disrespected, judged, excluded, marginalised,
>
> and even frightened.
>
> Racism and jealousy are rife.

Tomlins-Jahnk [87] states that it is not unusual for wahine Māori to work simultaneously in both worlds. Māori women are described as being initiators and the driving force for Māori. I am one of those initiators. I want to shift the institutional racism; how can I do that when there is only tokenistic gesticulation, never authentic and genuine actions towards Māori people and communities.

I have been judged, humiliated, and alienated because of my bodily parts. I don't have the privileges of my Pākeha DNA as my skin colour is brown. My mātauranga, te reo Māori and wairua have been used and abused, and my creativity around Te Ao has been marginalised and diminished. My Indigenous body was traumatised personally and professionally.

> I am not an administrator,
>
> I am not a secretary or a Māori advisor.
>
> I am a wahine Māori, a wife, a mother, a daughter,
>
> A warrior seeded and pollinated by my te uri o hau grandmother.

I now realise the trauma this has brought upon me as a young wahine Māori. I like to think of the two worlds in which I live like a tāniko finger weaving. There is a pattern known as aramoana (Figure 5) that represents pathways to many destinations by ocean and waterways, signifying growth and moving forward. My life weaves together many pathways and destinations; I (body, mind, spirit) am separate parts at times yet together at other times—a singular method, which in the case of tāniko, is a method to create a weaved piece. Creating a stronger, nurtured and directed mana wahine Māori.

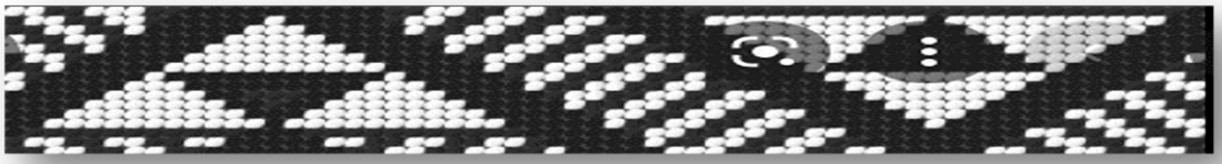

**Figure 5.** Tāniko finger weaving with aramoana pattern.

## 6. Discussion and Conclusions

The stories shared about Kaipara moana provide insights into the realities for Māori of the accumulated impacts of slow violence on the Indigenous body, nature and more-than-human worlds as a consequence of settler colonisation. We present the dimension of slow violence regarding the understanding of ecosystem degradation so that a just and ethical acknowledgement can be tendered. For the Māori authors of this paper, living amidst their ancestral territories is tainted by existing slow violence that has turned once vital and healthy land-/water-sea-scapes into sites of degradation and decline. For example, *why one scallop*, illustrates the (slow) violence applied to scallop kin that now has rendered it virtually extinct in the Kaipara. Such a demise continues to unfold and be felt at the intimate scale of the Indigenous body and household. The eco-poetic short story portrays the degradation of her more-than-human kin the scallop, detailing her profound concern for her scallop kin, the place she lives and the cultural responsibility of kaitiakitanga. The authors recognise the legacy of slow violence as it is evident when they (re)plant wetlands, forests and catchments; attend wananga (as described in Part 1), patrol rivers and monitor the mauri at polluted streams and coasts. The work was done to acknowledge both the grief and trauma of it all and celebrate the beauty of growing/becoming.

The eco-social violence inflicted upon Kaipara moana (and its human and more-than-human inhabitants) is traceable to settler colonial management structures that reinforced Western/scientific norms, values and understandings, and which simultaneously undermined and marginalised te ao Māori and all it encompasses. The stories *Tuna Saved my life, we survived off the Kaipara moana, section five this is bullshit*, and *te ao maori te ao pakeha* demonstrate ongoing resistance to the epistemic and slow violence of settler colonisation as well as the emotional (financial and personal) toll over multiple generations, that eco-social violence can impart on Indigenous peoples: the violence of sediment(ation) pollution, species and ecosystem loss, epistemic and ontological violence, the structural violence of institutional racism, capitalism, and globalisation as settler colonialism; and body violence. Rightly, we challenge environmental, fisheries, water and resource management policy to seek an in-depth understanding of ecological change through a lens of slow violence to privilege voices and bodies that are not frequently present in representations of land-/sea-/water-scapes. The Māori authors have spent most of their lives (over 20 years for some authors) resisting in cognitive, epistemic and practical ways. The practice of rahui, for example, detailed in *we survived off the Kaipara moana*, is a protocol and ritual, a *lore*, passed down over many generations and has been utilised by tribal elders to halt the (slow) violence planned for their more-than-human kin, the Kaipara moana. It has been used at times of great need when environmental law has been used to expedite such (slow) violence. Rahui was used to halt forestry plantations on fragile coastal land-scapes, to halt the placement of 200 underwater marine turbines; and (as detailed in *section five this is bullshit*) to halt population overfishing and destructive set-netting practices.

The stories can also be seen as a way of disrupting dominant (settler colonial) forms of knowledge production by showcasing different registers and personal (subjective) and embodied accounts of settler colonial (slow) violence. We present five narratives so Indigenous women and men can read and see themselves in eco-social violence, ecosystem degradation and sustainability discourse; and relate to the narratives, because it matters who is telling the story of ecosystem degradation and sustainable management. The exclusion of certain people's stories, knowledges, and histories from the mainstream (be it media

accounts, official histories, school curriculum, legal decisions, policies, and institutional arrangements) means they remain marginalised and vulnerable to future harm. Through storytelling, we sought to share the unseen violence exhibited on nature. For instance, Mina, Millan, Vicky, Mikaera, Mina and Alyssce provide intimate stories of eco-social violence and slow violence transcending generations, existing under the settler-colonial regime of environmental and resource management policy that produce(d)/maintain(ed) enduring inequalities between settler economic enterprises (such as industrial agriculture and forestry) and Indigenous peoples and their more-than-human kin.

The stories demonstrate how, for Indigenous peoples, settler colonialism is a form of slow (but hidden) violence experienced alongside their ancestors and their more-than-human kin (which includes land, water, plants, animals and spirits) who reside in their ancestral territories [58,88]. Such violence goes beyond the initial invasion and occupation of Indigenous territories and instead is ever-present and ongoing within Indigenous homes, families, and bodies. It is ever-present in sustainability management of ecosystems, shellfish such as scallop, fisheries, wetlands, and freshwater eel. Indeed, the work of De Leeuw, Bacon and Nixon [12,19,21,58,88] highlights the critical need for Indigenous and non-Indigenous ally researchers to examine how slow violence constrains Indigenous self-determination rights, impedes Indigenous capacities to access their lands/waters/foods, and impacts the intimate scales of home and the gendered body [54,88,89].

When scholars pay greater attention to the kōrero of Indigenous peoples, it becomes clear how deeply interwoven identity is with social-ecological connections and the harm that occurs when these relationships are overlooked or undermined. An important aspect of our research was how to grapple with the overlapping interplay of racism, violence, and ecosystem degradation across multiple generations and scales (temporal and spatial). Indeed, as we wrote about our lived experiences of pollution and the loss of food and its intersections with processes of marginalising tikanga (Māori laws), mātauranga (Indigenous knowledge), and relationships with more-than-humans, we were struck by how the micro-scale (Kaipara) was interwoven with the meso- (the neoliberal settler colonial state) and macro-scale (globalised political economies, colonialism, and neoliberalism). We were confronted by the macro realities of colonisation, globalisation and neoliberal agendas intersecting and damaging bodies, minds, and connections with our kin (humans and more-than-humans). The slow violence framing allowed us to conceptualise and work through the temporality of structural violence on both humans and more-than-humans; this is where the power of settler colonial violence lies: in and with time.

Illuminating slow violence remains immorally disquiet (unaddressed, unrecognized) in ecosystem (and/or sustainability) management. For us, this paper is a decolonising space. By highlighting our personal and collective kōrero, we drew attention to specific instances where we challenged the dominant ontological underpinnings of ecosystem-based management of land, water and sea as well as ecological restoration practices. In making seen/heard the unseen/unheard experiences of slow violence, we also challenge practices and conceptions of (post)sustainability to contemplate the significance of bringing deep, embodied and relational meaning to understand ecosystems at place, relations with nature and how to live with, in and beside nature.

We (Mina, Vicky, Leane, Alyssce, Millan, Mikaera) are markers of social difference present within settler-colonial regimes of environmental and resource management. The (re)framing of the natural world of Kaipara as a deep intimate relational more-than-human geography both makes visible slow violence perpetuated there, and also finds possibility for doing things otherwise. Our social life is more than relations between peoples, but with things, recognising that relations are always co-produced and co-constituted [65,66,68]. They are more-than-human encounters.

**Supplementary Materials:** The following supporting information can be downloaded at: https://www.mdpi.com/article/10.3390/su142214672/s1. Story: Tuna Saved My Life. By Millan Ruka; Story: Section Five [of the RMA]: "This is Bullshit" [69–74,76–86,90–94].

**Author Contributions:** Conceptualization, L.M.; Data curation, L.M.; Formal analysis, L.M.; Methodology, L.M., A.T.H., M.H., V.M., M.R. and M.M.; Project administration, L.M.; Supervision, M.P. and K.F.; Validation, A.T.H., M.H., V.M., M.R. and M.M.; Visualization, L.M., M.P., A.T.H., M.H., V.M. and M.R.; Writing—original draft, L.M., M.P., A.T.H., M.H., V.M., M.R. and M.M.; Writing—review & editing, L.M., M.P. and K.F. All authors have read and agreed to the published version of the manuscript.

**Funding:** This research received no external funding.

**Institutional Review Board Statement:** The study was conducted in accordance with the University of Auckland Human Participants Ethics Committee, reference number 018806.

**Informed Consent Statement:** Informed consent was obtained from all subjects involved in the study. This research has ethical approval from the University of Auckland Human Participants Ethics Committee 018806.

**Data Availability Statement:** Not applicable.

**Acknowledgments:** Leane wishes to acknowledge and thank Kaipara co-authors for sharing their stories and lived realities. The authors wish to acknowledge and thank the peer reviewers and the opportunity to share the story of the Kaipara moana in this special issue, Indigenous Transformations Towards Sustainability. Thank you, nga mihi nui.

**Conflicts of Interest:** The authors declare no conflict of interest.

## Abbreviations

| Glossary | Many translations sourced from https://maoridictionary.co.nz/, accessed on 1 January 2020. This website provides audio for the kupu/words. |
|---|---|
| Ahi kā | Burning fires of occupation |
| Aotearoa | Māori name for New Zealand |
| Aroha | To love, feel concern, affection |
| Awa | Waterway, river, stream |
| Hauora | Be fit, well, healthy, vigorous and in good spirits |
| Iwi | Tribal group |
| Kai | Food |
| Kaimahi | Worker, practitioner |
| Kai moana | Seafood, shellfish |
| Kaitiakitanga/kaitiaki | A social-environmental ethic that promotes a use agreement with natural ecosystems whereby an inter-generational and sustainable relationship between people and the ecosystem is retained within a customary area [95] (Kawharu. It is a contemporary "nurture and care" responsibility [85] which has intensified in response to the loss of biodiversity and the ecosystem degradation of ancestral land- and sea-scapes. A key imperative of kaitiakitanga is maintaining values of whakapapa, mana and mauri, health and vitality of ecosystems to protect their life-supporting properties. This role is performed by kaitiaki "a guardian, keeper, preserver, conservator, foster-parent, protector" [37] of places and things for the gods, and kaitiaki may not necessarily or are assumed to, take a human form. Kaitiakitanga is a practice that upholds tikanga [96] and has implications for health and wellbeing. Kaitiakitanga is political and concerned with Indigenous rights. |
| Kanohi ki te kanohi | Face to face |
| Karakia | prayer |
| Kaumatua | Adult, elderly man/woman, elder |
| Kaupapa | Project, programme, theme, issue, plan, matter for discussion |
| Kaupapa Māori | Is privileging Māori ontologies, knowledges and practices to research, learning, planning, health and language. In the research sector, includes the critique of colonialism and adversity alongside Māori agency and aspirations. Simply, means a Māori way of doing things; Māori approach, Māori agenda, Māori principles. |

| | |
|---|---|
| Kōtiro | Daughter, girl |
| Mahi | Work |
| **Mahinga kai** | Food gathering place |
| Mana | Prestige, authority, power, influence, status, spiritual power, charisma—mana is a supernatural force in a person, place or object. Mana goes hand in hand with tapu, one affecting the other. The more prestigious the event, person or object, the more it is surrounded by tapu and mana. |
| Mana whenua | Authority over tribal land or territory |
| Manaakitanga/manaaki | hospitality, reciprocity |
| Māori | Aboriginal inhabitant, Indigenous person, native |
| Marae | Courtyard in front of wharenui (meeting house) where formal greetings and discussions take place. Often used to include the complex of buildings around the marae. |
| Mātauranga | Indigenous wisdom, knowledge, knowing being and doing. Is philosophy, knowledge, method, values and language |
| Mate | Sick, ailing, unwell, diseased, be dead |
| Mauri | Life principle, internal energy or vital essence, source of emotions; derived from whakapapa, an essential essence or element sustaining all forms of life. Is the binding force that links the physical to the spiritual worlds (e.g., wairua). |
| Mihimihi | Greeting, formal speech, thank, pay tribute |
| Mimi | To urinate |
| Moana | Sea, ocean, coast, saltwaters |
| Pākehā | New Zealander of European descent |
| Papatūānuku | Primal parent, mother earth |
| Rahui | Restrictions on access and use of certain resources |
| Rohe | Tribal land, waters and area |
| Te ao Māori | Māori ontology |
| Taonga | Treasure |
| Tapu | Sacred |
| Tuna | Eel. The longfin eel, *Anguilla dieffenbachi*, (conservation status: endangered) are diminishing from loss of habitat, and suitable water quality and lack of suitable access for kaitiaki to manaaki tuna [97]. |
| Te taiao | Ecosystem, environment, nature |
| Tikanga | Māori lore/law, correct procedure, custom, practice |
| Turangawaewae | Standing, place where one has the right to stand, place where one has rights of residence and belonging through kinship and whakapapa |
| Wāhi tapu | Forbidden, sacred |
| Wahine | Female essence |
| Wai | Water, stream, creek, tears |
| Wairua | Spirit, spirituality |
| Waka | Canoe, vehicle, traditional Māori canoe |
| Whakapapa | Genealogy |
| Whenua | Land, placenta |

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
