# Peer review of "(Un)Heard Voices of Ecosystem Degradation: Stories from the Nexus of Settler-Colonialism and Slow Violence"

_sustainability, doi:10.3390/su142214672_

Round 1

Reviewer 1 Report

I suggest you to use the dialect words between bracket. I believe it would be easier for the reader to understand better the message.

Also, present the bias effect due the fact that the authors tell a part of their experiences. And also the limits of the study.

Kind regards,

Reviewer 2 Report

- Location/region in the title could be made clearer with some punctuation or rephrasing. Also there should be a question mark as it stands, although I find the last phrase in the title unnecessary.

- Very important topic developing important and interesting concepts.

- Could do with being proof-read/copyedited by a professional.

- It's extremely thorough but too dense and long, and one should be careful to make the key points and arguments stand out from the storytelling, which is of course important too. I would try to make it as concise as possible, following the structure and conceptual thread as presented in the abstract. I would try to reduce by up to half.

- Get lost in the examples and details a bit.

- I wouldn't use 'I' in a multi-author paper unless identifying who is writing. 

- I think the paper, especially the second half, suffers from lack of cohesion and concision. Try to get it to half the length with the main conceptual steps and arguments, and only give details and examples which are absolutely relevant to these mains threads.

- Generally I think the paper looks good until sections 4 and 5, where it then loses its way. These sections ('Materials and Methods' and 'Results', although I don't really know why they're called that) - especially the second - seem thrown in and do not really connect or flow well with what proceeded, and are exceedingly long. I would cut these sections drastically, and anything you do keep merge into the flow and style of the preceding sections more. This could give the opportunity to make the arguments and conclusions stronger too.

Reviewer 3 Report

This is an excellent and valuable long paper detailing the cultural and environmental destruction of Kaipara moana. It provides a powerful example of the colonising process and all the violence that involves (present tense) against Maori and their non-human kin. The telling of this history examines the incompatibility of western scientific discourse in conversation with traditional ecological knowledge systems, and also documents the inequity of the power systems involved. Understanding these systems, both Pakeha and Maori, is crucial for the future if there is to be reparation and acknowledgment of harm done. 

The methodology is explained well and forms a strong framework for the report, and for the important personal accounts documented in the study.

In documenting the "slowly unfolding environmental catastrophes" Aotearoa appears like a microcosm of global monoculture - but now operating at break-neck speed, while the initial impacts were relentless, slow and deep. The increase in the use of nitrogen fertilisers by 629% in 28 years is criminal by any global measure, and catastrophic, highlighting the systemic corruption and violence within NZ that is out of control. This one figure alone signals a complete regulatory failure on the part of the government.

On the general text, it is well written and informative. There are a few small corrections and queries that need to be fixed/clarified in the final copy but these are only minor adjustments.

The year 925AD is listed as the arrival of the Maori to NZ, is there a reference for this date - usually it is quoted as much later, around 1280, so please clarify.

line 213 necessary not necessarily

The diagram on Page 8 and 9 needs checking, the copy has mistakes:

Under Commercialisation of oysters line three:

I think this should read “kauri timber industry” not time, (and lower case k)

Under “Overfishing and Species Decline”

The first line (or word) is missing. Check last sentence re-oysters: perhaps “oysters growing with in the area were reported as tasting muddy, covered with slimy mud, or killed by mud.”

Under Aquaculture

“Inshore fish populations” …typo

Under Species Loss

last line typo “remaining"

Page 9

under Radical Chances

Repetitive: wetlands drained x 2, take out second and start with “Non-exotic trees”…

“Fish () were netted by hand (not was netted). Native rock oyster were …

Check order of the figures, I think the one on page 9 should come first.

Figure 3 “Matauranga” Check last sentence, take out “Taken”

Line 267 “a state of mate”: – can you define mate in brackets for clarity, is that the correct term here?

Line 270 dominate or dominance (not dominant)

Lines 279-280 capable of being owned, that was …. (add comma)

Line 287 were seeded

Line 414 Councillor (not Council)

P. 14 footnote 3, could you please put in the scientific names of the species referred to? It is a bit unclear if it is referring to fish or eel.

Line 429 Formating: would it be clearer to separate the Maori from the English? (separate paragraphs)

Line 438: river, land, gods or biota – should that be “and biota”?

Line 451 should this say “Well, now you can’t see anything” – the past tense is confusing (couldn’t see) but perhaps that is the way it was said?

Line 699 “[We] were [are] starving right on the shore,” says Mikaera. Repeats again line 718 is this intentional?

Line 1128 reference is that page no correct?

Supplementary materials

Millan Ruka’s account includes details about the speed that dairy took over, this is an important paragraph  that perhaps could go in the main copy. 

There is a reference to Figure XX in the supplementary material on the “Funnel”, what is this refereeing to?

Many thanks to the authors for this paper.

Round 2

Reviewer 2 Report

I do not see any significant changes in content, so my first review still stands in this regard. 

Author Response

please see attached letter
